# Immunogenicity, Safety, and Immune Persistence of One Dose of SARS-CoV-2 Recombinant Adenovirus Type-5 Vectored Vaccine in Children and Adolescents Aged 6–17 Years: An Immunobridging Trial

**DOI:** 10.3390/vaccines12060683

**Published:** 2024-06-19

**Authors:** Xu Han, Mingwei Wei, Xiuyu Zheng, Peng Wan, Jie Tang, Lu Zhang, Shupeng Zhang, Hanchi Zhou, Jiayu Lu, Li Zhou, Yawen Zhu, Jingxin Li, Fengcai Zhu

**Affiliations:** 1School of Public Health, National Vaccine Innovation Platform, Nanjing Medical University, Nanjing 211166, China; hanxu930217@stu.njmu.edu.cn (X.H.); zhouli293020000@163.com (L.Z.); zhuyawen1998zyw@163.com (Y.Z.); 2Jiangsu Provincial Medical Innovation Center, National Health Commission Key Laboratory of Enteric Pathogenic Microbiology, Jiangsu Provincial Center for Disease Control and Prevention, Nanjing 210009, China; js_wmw@163.com; 3Cansino Biologics Inc., Tianjin 300457, China; xiuyu.zheng@cansinotech.com (X.Z.); peng.wan@cansinotech.com (P.W.); lu2.zhang@cansinotech.com (L.Z.); shupeng.zhang@cansinotech.com (S.Z.); hanchi.zhou@cansinotech.com (H.Z.); 4Funing Country Center for Disease Control and Prevention, Yancheng 224435 China; 13962077131@163.com; 5Shanghai Jiao Tong University Affiliated High School IB Curriculum Center, Shanghai 200439, China; lujiayu2020@outlook.com

**Keywords:** immunobridging trial, batch-to-batch consistency, Ad5-nCov, clinical trial, children and adolescents

## Abstract

Background: Though children infected by SARS-CoV-2 generally experience milder symptoms compared to adults, severe cases can occur. Additionally, children can transmit the virus to others. Therefore, the availability of safe and effective COVID-19 vaccines for children and adolescents is crucial. Method: A single-center, randomized, double-blind clinical trial was conducted in Funing County, Yancheng City, Jiangsu Province, China. Healthy children and adolescents were divided into two subgroups (6–12 years old or 13–17 years old) and randomly assigned to one of three groups to receive one dose of Ad5-nCoV (3 × 10^10^ vp/dose). Another group, aged 18–59, received one dose of Ad5-nCoV (5 × 10^10^ vp/dose) as the control group. At 28, 90, 180, and 360 days post-vaccination, we measured the geometric mean titer (GMT)/concentration (GMC) of neutralizing and binding antibodies against the prototype SARS-CoV-2 strain, as well as serum antibody levels against the BA.4/5 variant. We also evaluated the incidence of adverse events within 28 days post-vaccination. Results: A total of 2413 individuals were screened from 3 June 2021 to 25 July 2021, of whom 2021 eligible participants were enrolled, including 1009 aged 6~17 years in the children and adolescent group and 1012 aged 18–59 years in the adults group. The GMT of anti-wild SARS-CoV-2 neutralizing antibodies was 18.6 (95% CI, 16.6–20.9) in children and adolescents and 13.2 (95% CI, 11.6–15.0) in adults on day 28. The incidence of solicited adverse reactions between the adult group (49.4% [124/251]) and the children and adolescent group (46.3% [156/337]) was not statistically significant. The neutralizing antibody levels decreased by a factor of 3.29 from day 28 to day 360 post-vaccination. Conclusions: A single dose of Ad5-nCoV at 3 × 10^10^ virus particles/dose is safe in children and adolescents, and it elicited significant immune response, which was not only non-inferior but also superior to that in adults aged 18–59 years.

## 1. Introduction

The COVID-19 pandemic has imposed an enormous burden on public health, socio-economic factors, and livelihoods in the past 3 years [1]. SARS-COV-2 can infect people of all ages, and although COVID-19 is usually milder in children than in adults, it may still lead to severe illness and long-term complications after infection, including multi-system inflammatory syndrome (MIS-C) in children [2,3,4,5]. Although increasing numbers of COVID-19 vaccines are now being authorized for use in adults aged 18 years and above, only a few are recommended for use in children [6]. BNT162b2 was approved on the WHO Emergency Use List (EUL) for people 5 years of age and older. On 2 November 2022, the World Health Organization (WHO) approved the extension of the age of application of Sinovac-CoronaVac to 3 years of age, the first COVID-19 vaccine on the EUL for 3 years of age [7].

The COVID-19 vaccine CONVIDECIA, developed by CanSino Biologics, is a non-replicating Ad5 adenovirus vector COVID-19 vaccine (AD5-NCoV). On 19 May 2022, the World Health Organization (WHO) announced CONVIDECIA’s inclusion in the “Emergency Use List” (EUL) for adults aged 18 years and above [8]. A phase 2b trial of CONVIDECIA showed that a single dose of the Ad5 vector COVID-19 vaccine was safe and induced a robust immune response in children and adolescents aged 6–17 years, with ELISA-RBD IgG levels decreasing with increasing age [9].

However, due to limited research on Ad5-nCoV in children, the vaccine is currently only recommended by the WHO for use in people over 18 years of age. This study verified through an age immunobridge trial that the immunogenicity and safety of Ad5-nCoV in the 6–17 age group are non-inferior to that of the 18–59 age group. We also demonstrated the consistency of three batches of Ad5-nCoV in the 6–17 age group and immune persistence of Ad5-nCoV in this age group.

## 2. Method

### 2.1. Study Design

This was a single-center, randomized, double-blind clinical trial conducted in Funing County, Yancheng City, Jiangsu Province, China. The immunogenicity, safety, and immune persistence of a single intramuscular dose of Ad5-nCoV vaccine were evaluated in children and adolescents aged 6–17 years, as well as the consistency of vaccine immunogenicity between different batches. The study hypothesizes that the Ad5-nCoV vaccine is safe and can induce SARS-CoV-2 neutralizing antibodies in children and adolescents aged 6–17 years, with a comparable or potentially superior immune response compared to adults aged 18–59 years, and with durability. The protocol and informed consent were reviewed and approved by the Ethics Committee of the Jiangsu Center for Disease Control and Prevention (JSJK2021-A010-01) before the launch of the trial, and no changes were made to the protocol after the study began. Before enrolling in the study cohort, the written informed consent forms were obtained from participants equal to or over 18 years of age and children or adolescents 6–17 years of age as well as their guardians. This trial was registered with ClinicalTrials.gov (NCT04916886) and undertaken in compliance with the principles of the Declaration of Helsinki, ICH-Good Clinical Practice guidelines, and local guidelines.

### 2.2. Participants

Healthy participants, male or female, 6–17 years of age and 18–59 years of age who had not received any previous dose of a COVID-19 vaccine, were recruited for eligibility screening by the local Center for Disease Control and Prevention. The investigators examined the participants’ vaccination records and medical histories. The major criteria for inclusion were axillary temperature ≤37 °C, negative anti-SARS-CoV-2 IgM and IgG in serum, and previous medical history and physical examination indicating a healthy condition. Exclusion criteria include a history of SARS-CoV-2 infection or COVID-19 epidemiological exposure, a previous vaccination against COVID-19, a severe acute allergic reaction to the any ingredient of vaccine, and other serious medical conditions (Appendix A for complete inclusion and exclusion criteria).

### 2.3. Randomization and Masking

The eligible participants aged 6–17 years were stratified to two age subgroups, 6–12 and 13–17 years old, and then were randomly assigned at a ratio of 1:1:1 to receive one of the three batches of the Ad5-nCoV (NCOV202104001V, NCOV202104002V, and NCOV202104004V) vaccines through the utilization of interactive web-based response randomization system (IWRS). Participants aged 18–59 who were assigned to receive batch NCOV202104001V were considered as the reference group. Subsequently, the first 120 individuals from the 6–17 age group were included for the investigation of immune persistence. Informed non-blinded personnel assigned to administer vaccines were aware of batch group allocations but were prohibited from participating in any other research processes or disclosing vaccine administration and grouping details to other researchers or trial participants. The random list was generated by an independent statistician using SAS (version 9.4) and imported into IWRS. Other researchers, lab staff, and participants were all masked to the batch group allocation.

### 2.4. Interventions

Ad5-nCoV is a replicating defective Ad5-vector vaccine jointly produced by China Institute of Biotechnology (Beijing, China) and CanSino Biologics (Tianjin, China), which expresses the full-length spiking glycoprotein gene of prototype SARS-CoV-2 (Wuhan-Hu-1). It is delivered in liquid form in a pre-filled syringe containing 5 × 10^10^ virus particles/vial (0.5 mL). The recommended vaccination dosage is 0.5 mL for adults aged 18–59 and 0.3 mL for children and adolescents aged 6–17. Vaccines were stored at 2–8 °C and administered as a single dose of Ad5-nCoV into the deltoid muscle.

### 2.5. Study Procedures

All participants were observed for any vaccine-related adverse reactions in the outpatient clinic for at least 30 min. They were then instructed to self-record any solicited or unsolicited adverse reactions on a diary card daily for the next 28 days, either by themselves or with the help of a guardian. Solicited injection site adverse events include pain, swelling, induration, itchiness, and erythema. Solicited systemic adverse events include hypersensitivity, fever, fatigue, and headaches, among others. The severity of adverse events was graded according to the China State Food and Drug Administration (2019 edition) (e.g., fatigue or weakness, Grade 1: Does not affect daily activities; Grade 2: Affects normal daily activities; Grade 3: Severely affects daily activities, unable to work; Grade 4: Emergency or hospitalization) [10], and the causality with vaccination was judged by investigators before unmasking (until the end of the study unblinding (except for meeting emergency unblinding conditions), all subject group information and drug trial information were not to be disclosed). Severe adverse events were self-reported by participants or their guardians. Blood samples were collected from all participants before vaccination and on day 28 post-vaccination. A subset of 120 participants was selected for the immune persistence study, with blood samples collected at 90, 180, and 360 days post-vaccination.

### 2.6. Outcomes

The primary endpoint of immunogenicity was the geometric mean titer (GMT) of the SARS-CoV-2 neutralizing antibody against the prototype live virus. The SARS-CoV-2 neutralizing antibodies were measured by the microneutralization method based on cytopathic effect using a SARS-CoV-2 wild-type strain BetaCoV/Jiangsu/JS02/human/2020A (GISAID EPI_ISL_411952). The reported antibody titer is the inverse of the highest sample dilution that protects at least 50% of cells from cytopathic effects. The serum was diluted in a two-fold gradient, ranging from 1:4 to 1:256. The secondary immunogenicity endpoints included SARS-CoV-2 RBD-specific IgG antibodies (measured using a commercial anti-SARS-CoV-2 RBD-IgG ELISA kit from Vazyme Medical Technology, Nanjing, China) and Spike (Omicron BA.4/5)-specific IgG antibody responses (measured using a commercial anti-Spike (Omicron BA.4/5)-IgG ELISA kit from Vazyme Medical Technology, China). The cutoff value of ELISA antibody concentration was 20 BAU/mL or 6.4 BAU/mL. Pre-existing anti-Ad5 neutralizing antibody in the serum at baseline before the vaccination was also measured by a neutralization assay as previously mentioned [11]. The primary safety endpoints were the incidence of solicited and unsolicited adverse reactions within 28 days of vaccination, and the incidence of adverse reactions within 30 min and 7 days.

### 2.7. Statistical Analysis

The sample size calculation was conducted based on the immunogenicity data from the previous clinical trial (the GMT of neutralizing antibodies on day 28 was 18.4 with a standard deviation (SD) of 3.8), and the coefficient of variation for GMT was estimated to be 2.2. Assuming that the GMT of the 6–17 age group was non-inferior to that of the 18–59 age group, with the adjusted one-sided significance level of 0.025, GMT ratio = 0.67, and a 1:1 group ratio, the calculated minimal sample size for each group was 228 in order to achieve a power of 0.90. Taking into account a dropout rate of 10%, we determined that a sample size of 253 was required for the non-inferiority hypothesis. Then, we also estimated the sample size for the three batches’ consistency tests in the children and adolescent groups. Assuming that the immunogenicity results of the vaccine for three batches are equivalent, the GMT ratio of any pair of batches is within the range of 0.67–1.5. With a corrected two-sided significance level of α = 0.017, and an equal sample size for each batch group, the calculated sample size for each of the three batches was 314, in order to achieve a power of 0.90. Considering a dropout rate, the final determined sample size was 336 individuals for each batch. 

Therefore, the study included 1008 participants in the children and adolescents age group (with 336 individuals in each batch group), with an equal number of individuals in two age subgroups (6–12 years old and 13–17 years old). Additionally, the number of individuals in each subgroup was no less than 500, meeting the safety evaluation requirements. Additionally, 253 subjects were included in the adult age group (18–59 years old).

Antibody responses were calculated with the log-transferred neutralizing antibody titer and are reported as the GMT with 95% CI. When GMT was calculated, the SARS-CoV-2 neutralizing antibodies lower than the minimum detection limit were input with half of the threshold. If the baseline titer was undetectable but antibodies were detectable after vaccination, or if the antibody level increased by at least fourfold compared to the detectable baseline level, participants could be defined as having seroconverted. Non-inferiority was defined as the lower limit of the two-sided 95% confidence interval of a GMR > 0.67. The Shapiro–Wilk test was used to test whether the sample is normally distributed, the Wilcoxon rank sum test was used to compare non-normal distribution data, and ANOVA or *t* tests were used to compare normal distribution data. Subgroup analysis was performed based on the pre-existing anti-Ad5 neutralizing antibody levels (titers ≤1:200 or >1:200). Safety analyses stratified by age were presented on day 28 post-vaccination. Categorical data were tested using the chi-square test or Fisher’s exact test for statistical analysis. The influence factors of SARS-CoV-2 neutralizing antibodies were analyzed by multiple linear regression. Statistical tests were analyzed using SAS 9.4 or GraphPad Prism 8.0.1.

## 3. Results

### 3.1. Profiles of Clinical Trials and Demographic Characteristics of Participants

A total of 1071 children and adolescents 6–17 years of age were screened for eligibility from 12 July 2021 to 25 July 2021. Of them, 1009 eligible participants were recruited with 505 enrolled in the 6–12 years cohort and 504 in 13–17 years cohort, and then they were randomly assigned to receive one of three batches (NCOV202104001V, NCOV202104002V, and NCOV202104004V) of Ad5-nCoV at a 0.3 mL/dose. A total of 337 subjects aged 6–17 years received one dose of Ad5-nCoV (batch NCOV202104001V) as the experimental group in the immunobridging study (Figure 1). In the meantime, a total of 1342 adults 18–59 years of age were screened for eligibility from 3 June 2021 to 7 June 2021, and 1012 eligible participants were recruited. Of them, 253 participants were randomly assigned to receive batch NCOV202104001V of Ad5-nCoV at a 0.5 mL/dose as a control group in this study (Figure 1). We selected the first 120 enrolled subjects aged 6–17 years for the study on immune persistence (Figure 1).

A total of 1008 people in the 6–17 age group were included in the safety analysis (1 was lost to follow-up), and 1001 (99.3%) participants who donated blood samples on day 28 after vaccination were included in the immunogenicity analysis. The baseline demographic characteristics of participants are shown in Table 1. Baseline data of three batches of participants aged 6–17 years are comparable.

### 3.2. Age Bridging Trial

#### 3.2.1. Immunogenicity

Before vaccination, the SARS-CoV-2 neutralizing antibodies were not detectable in all participants. After 28 days of vaccination, neutralizing antibody GMT increased to 18.6 (95% CI, 16.6–20.9) in children and adolescents and to 13.2 (95% CI, 11.6–15.0) in adults, resulting in a GMT ratio of 1.41 (95% CI 1.15–1.72) for children and adolescents versus adults (Figure 2B). The vaccine elicited SARS-CoV-2 neutralizing antibodies in the child and adolescent cohorts that were not only non-inferior to that in the adult cohort but also superior to that in the adult cohort (*p* < 0.0001). Moreover, 28 days after vaccination, the GMT of neutralizing antibodies increased to 22.7 (95% CI, 19.2–26.9) in children, significantly higher than 15.2 (95% CI, 13.1–17.7) in adolescents (Figure 2D).

Similarly, after 28 days of vaccination, RBD-specific binding antibodies were higher in children and adolescents (179.4 [95% CI, 160.0–201.3]) than in adults (77.0 [95% CI, 66.9–88.6]) (*p* < 0.0001) (Figure 2A). And we found that 28 days after vaccination, the GMC of binding antibodies in children increased to 257.6 (95% CI, 222.3–298.5), significantly higher than 124.7 (95% CI, 106.2–146.4) in adolescents (Figure 2C).

In terms of the seroconversion of neutralizing antibodies, 92.24% of participants aged 6–17 years and 91.8% of participants aged 18–59 years had seroconversion after 28 days of vaccination, with no significant difference between the two groups (*p* = 0.848). Meanwhile, for the seroconversion of anti-RBD IgG, 97.3% of subjects in the 6–17 years group and 88.5% in the 18–59 years group were observed (*p* = 0.064) (Figure 2E).

The levels of pre-existing anti-Ad5 neutralizing antibodies in the two age cohorts were similar 100.46 (77.09–130.91) vs. 140.60 (111.17–177.41) (*p* = 0.073) (Appendix A). Then, subgroup analysis was performed according to the level of pre-existing anti-Ad5 antibodies; when the GMT of anti-Ad5 neutralizing antibodies at baseline was >200, there was no difference in the SARS-CoV-2 neutralizing antibody GMT between the two age groups (*p* = 0.2003). When the baseline anti-Ad5 neutralizing antibody GMT was ≤200, the SARS-CoV-2 neutralizing antibody GMT in the 6–17-year-old group was significantly higher than that in the 18–59-year-old group (*p* < 0.001) (Appendix A). Moreover, we found that the neutralizing antibody GMT of SARS-COV-2 live virus was lower in groups with high titers of pre-existing anti-Ad5 antibodies in all age groups (*p* < 0.001) (Appendix A).

#### 3.2.2. Safety

Within 28 days of vaccination, there was no significant difference in the incidence of solicited adverse reactions between the adult group (49.4% [124/251]) and the children and adolescent group (46.3% [156/337]) (*p* = 0.455) (Figure 3). The incidence rates of grade 3 solicited adverse reactions or unsolicited adverse reactions between the children and adolescent group and adult group was not statistically significant, at 3.26% versus 2.79% (*p* = 0.741) or 0.29% versus 0.39% (*p* = 0.834), respectively (Figure 3). Pain, fatigue, headache, and fever were the most common adverse reactions. Most adverse reactions were mild or moderate and generally resolved within 7 days. However, the incidence of systemic fever and vomiting in the 6–17-year-old group (31.8%, 107/337; 3.0%, 10/337) was statistically higher than that in adults (12.4%, 31/251; 0.4%, 1/251) with *p* < 0.05. In contrast, the incidence of fatigue and injection site pain in the 18–59 age group (12.0%, 30/251; 28.7%, 72/251) was statistically higher than that in the 6–17-year-old group (7.1%, 24/337; 16.9%, 57/337) with *p* < 0.05. In terms of unsolicited adverse reactions, there was no statistical significance between the two age cohorts according to systematic organ classification. No serious adverse events occurred during the 28-day follow-up period.

### 3.3. Batch-to-Batch Consistency

On day 28 after vaccination, the SARS-CoV-2 neutralizing antibody GMT values were 18.6 (95% CI, 16.6–20.9), 17.2 (95% CI, 15.2–19.4), and 19.9 (95% CI, 17.6–22.4) in the three batch groups, respectively, in the 6–17 age cohort (Appendix A). The immune responses of the three batches of Ad5-nCoV were equivalent on day 28 with 95% CI for GMT ratios between 0.67 and 1.5 for pairwise comparisons between the different batches (Figure 4). A single dose of Ad5-nCoV caused similar seroconversions of neutralizing antibodies in 91.0–92.5% of participants in three groups at 28 days after vaccination (Appendix A). Before vaccination, 554 of 1008 participants (55%) had high levels of anti-Ad5 neutralizing antibodies (Table 1). Multiple linear regression was performed for the SARS-CoV-2 neutralizing antibodies. We found that high pre-existing Ad5 immunity (titer of >1:200 vs. ≤1:200) was associated with a significantly lower level of SARS-CoV-2 neutralizing antibodies (β = −0.225, 95% CI, −0.25, −0.20) (Figure 6A). However, differences in the Ad5-nCoV batch were not the influencing factors for neutralizing antibody levels.

Similarly, the GMC of SARS-CoV-2 RBD-specific IgG showed similar results. The immune response of three batches of Ad5-nCoV on day 28 was equivalent in the 6–17 age group (Figure 5; Appendix A). The level of pre-existing Ad5 neutralizing antibodies is what affects SARS-CoV-2 RBD-specific IgG, while different batches do not (Figure 6B).

Comparisons of local, systemic adverse reactions, and adverse reactions incidence among batches showed no statistically significant differences within 28 days after the vaccination (Appendix A). Furthermore, subgroup analysis of adverse reactions within 7 days after the vaccination was conducted according to the 6–12 age group and the 13–17 age group. Detailed results are shown in Appendix A.

### 3.4. Immune Persistence

We randomly selected the first 120 individuals from the 6–17 age cohort for a longitudinal study on the persistence of antibody levels. Finally, a total of 100 individuals were enrolled in this study. Following the injection of Ad5-nCoV, the SARS-CoV-2 prototype RBD binding antibodies’ GMC peaked at 177.8 BAU/mL (95% CI, 144.9–218.2) on day 28, gradually decreasing thereafter. At 90 days, 180 days, and 360 days post-vaccination, the GMC values were 67.7 BAU/mL (95% CI, 49.7–91.9), 31.9 BAU/mL (95% CI, 24.6–41.3), and 23.7 BAU/mL (95% CI, 19.2–29.2), respectively. There was a notable 7.48-fold decrease from peak to trough (Figure 7). Similarly, the SARS-CoV-2 neutralizing antibodies’ GMT values at 28 days, 90 days, 180 days, and 360 days post-vaccination were 23.3 (95% CI, 18.9–28.5), 15.7 (95% CI, 11.9–20.5), 9.3 (95% CI, 7.1–12.3), and 7.1 (95% CI, 5.4–9.2), respectively. This represented a 3.29-fold decrease from peak to trough (Figure 7). Furthermore, we conducted quantitative detection of binding antibodies against Spike (Omicron BA.4/5) (Figure 8). The results showed that the levels of binding antibodies at 28 days, 90 days, and 180 days post-vaccination were 57.7 BAU/mL (95% CI, 48.5–68.8), 20.3 BAU/mL (95% CI, 15.9–25.8), and 6.9 BAU/mL (95% CI, 5.8–8.4), respectively. The antibody levels decreased by 8.27-fold from the peak level at 28 days to 180 days.

## 4. Discussion

Due to safety and vaccine priorities, most COVID-19 vaccines do not initially include children and adolescents as participants in clinical trials. At present, there are not many COVID-19 vaccines available for use in children or adolescents that are included in the WHO Emergency Use List [7]. COVOVAX and Nuvaxovid are for people over 12 years old, mRNA-1273 is for people over 6 years old, and BNT162b2 is for people over 5 years old [7]. The COVID-19 inactivated vaccine (BBIBP-CorV), developed by the Beijing Institute of Biological Products and the Chinese state-owned company Sinopharm, can be used in the population aged 3 and above [7]. The COVID-19 inactivated vaccine (PiCoVacc) produced by Sinovac Biotech can be used in the population aged 3–59 [7]. Whether a vaccine is suitable for people mainly depends on two aspects: one is the safety of the vaccine, and the other is the effectiveness of the vaccine. The study indicates that mRNA-1273 has been approved for adolescents and children aged 6 months and above, with lower doses (50 µg for ages 6–11 and 25 µg for ages 6 months to 5 years), showing over 90% efficacy against infection [12]. BNT162b2 was also used for adolescents and children aged 6 months and above, with lower doses (10 µg for ages 5–11 and 3 µg for ages 6 months to 4 years, compared to the standard 30 µg dose), demonstrating over 90% efficacy against infection [12]. Mid-term analysis of Phase III clinical trials in Turkey (NCT04582344) and Indonesia shows that the COVID-19 inactivated vaccine (PiCoVacc) has efficacy rates of 83.5% and 65.3% against symptomatic infection, respectively [12]. The above results indicate that COVID-19 vaccines on the World Health Organization’s Emergency Use List are generally effective. However, adverse events following immunization (AEFIs), especially for adverse events of special interest (AESIs), should be paid more attention. A study on COVID-19 in children indicated that in national surveillance data, the combined reporting rate of AEFIs for the mRNA vaccine BNT162b2 produced by BioNTech and Pfizer was 3424.5 per million doses (95% CI, 2725.7–4123.3), and for the inactivated vaccine BBIBP-CorV produced by the Beijing Institute of Biological Products and the state-owned enterprise Sinopharm Group, the combined reporting rate of AEFIs was 316.4 per million doses (95% CI, 285.8–347.0) [13]. In clinical trials of COVID-19 inactivated vaccine BBIBP-CorV (Beijing Institute of Biological Products Co., Ltd., Beijing, China), AEFIs were less common in children or adolescents than in adults, but mRNA vaccine BNT162b2 showed the opposite result [13]. And as reported in a previous article, BNT162b2 was found to be associated with a higher risk of myocarditis in children and adolescents [14]. Therefore, the safety and immunogenicity of the COVID-19 adenovirus vector vaccine (Ad5-nCoV) used in children and adolescents to prevent infection from COVID-19 remains an important question.

In this single-center, randomized, double-blind clinical study, we compared the safety and immunogenicity of a single-dose Ad5-nCoV vaccine in children and adolescents with that in adults. The results of our study showed that the Ad5-nCoV vaccine had superiority in terms of GMT in children and adolescents compared with that in adults, with a GMT ratio of 1.41. Similar trends of enhanced immune responses associated with decreasing age have been observed in other COVID-19 vaccines. Both the inactivated SARS-CoV-2 vaccine (CoronaVac) and BNT162b2 exhibit higher antibody titers in children and adolescents compared to adults and the elderly [15,16]. This finding may be due to the weight dependence of the vaccine, and similar speculation has been put forward in one study about the BBIBP-CorV vaccine [17]. The Phase 3 trial of the Ad5-nCoV vaccine showed an efficacy of 57.5% in participants aged 18 years or older [18]. Age immunobridging trials suggest that the immunogenicity in the children and adolescent group is superior to that in the adult group, indicating that the Ad5-nCoV vaccine may have comparable or even better protective efficacy in the 6–17 years group. Further, no difference was found between the two age groups in the pre-existing anti-Ad5 antibody. When the titer of the anti-Ad5 antibody was > 200, there was no difference in the SARS-CoV-2 neutralizing antibodies’ titer between the two groups on day 28 (*p* = 0.2003). When the anti-Ad5 antibody titer was ≤ 200, the SARS-CoV-2 neutralizing antibodies’ titer in the children and adolescent group was significantly higher than that in the adult group on day 28 (*p* < 0.001). And we found that at any age, pre-existing Ad5 immunity could slow down the rapid immune responses to SARS-CoV-2, which is consistent with previous research findings [19].

In addition, we compared the safety and immunogenicity consistency of three batches of Ad5-nCoV vaccines in the children and adolescent group. On day 28 after vaccination, the immunogenicity of both RBD-IgG binding antibodies and SARS-CoV-2 (index) neutralizing antibodies were well demonstrated, confirming the reliability and robustness of the production process. At the same time, multiple linear regression was performed for the SARS-CoV-2 neutralizing antibodies and RBD-specific IgG. We found that high pre-existing Ad5 immunity (titer of >1:200 vs. ≤1:200) was associated with a significantly lower level of antibody response. At the same time, different batches did not cause changes in antibodies response. We speculate that pre-existing anti-Ad5 antibodies may reduce the immune response to SARS-CoV-2 infection by neutralizing the vaccine vector and affecting the efficiency of vaccine particle entry into cells. Despite the ubiquity of adenovirus in children, the average levels of SARS-CoV-2 antibodies induced by Ad5-nCoV in the 6–17 age group remain higher than in the 18–59 adult group.

The Ad5-nCoV vaccine had a good safety, and adverse events within 28 days were generally tolerated and subsided within a short time in the children and adolescent group and adult group. Some solicited adverse reactions (such as fever, headache, erythema, and pain, induration, and swelling at the injection site) were reported as grade 3. However, there were no significant differences in the incidence of grade 3 solicited adverse reactions and grade 3 unsolicited adverse reactions between the children and adolescent group and adult group, with rates of 3.26% versus 2.79% (*p* = 0.741) and 0.29% versus 0.39% (*p* = 0.834), respectively. In the children and adolescent group, the most common systemic reactions were fever, headache, and fatigue, and the most common local adverse reactions were pain, swelling, and induration at the injection site. We also found no difference in the incidence of any solicited adverse reactions within 28 days between the children and adolescent group and the adult group. The 6–17 age group had a higher incidence of fever (31.8% vs. 12.4%, *p* < 0.0001) and vomiting (3.0% vs. 0.4%, *p* = 0.023), while the 18–59 age group had a higher incidence of fatigue (12.0% vs. 7.1%, *p* = 0.045) and pain (28.7% vs. 16.9%, *p* = 0.001) at the injection site. The difference in incidence of adverse reactions may be due to differences in immune system development between these two populations. Furthermore, there was no difference in the incidence of adverse reactions within 28 days among different batches of Ad5-nCoV vaccines in the 6–17 age group. While no serious adverse events were observed in children and adolescents in this study, the safety of Ad5-nCoV still requires continuous monitoring and evaluation in the future.

In this study, the immune durability of Ad5-nCoV was also investigated. The study revealed that the levels of SARS-CoV-2 neutralizing antibodies decreased from a peak of 23.3 (95% CI, 18.9–28.5) at 28 days to 9.3 (95% CI, 7.1–12.3) at 180 days, representing a 60% decline. We observed that the neutralizing antibody level induced by two doses of the mRNA vaccine BNT162b2 in adults was 42.26, and after 180 days post-vaccination, the neutralizing antibody geometric mean titer (GMT) decreased to 25.11, a decline of 40.6% [20]. Although the GMT of neutralizing antibodies induced by Ad5-nCoV is not as high as that induced by BNT162b2 and shows a larger decline, it is challenging to determine which vaccine demonstrates superior durability due to BNT162b2 being administered in two doses and the studies being conducted in different laboratories. Another study conducted by our research group under the same conditions regarding the durability of a single dose of BNT162b2 seems to provide a better comparison. One month after vaccination, the GMT of neutralizing antibodies increased to 324.35 (305.14–344.77), and after six months post-vaccination, the GMT of neutralizing antibodies decreased to 20.35 (19.12–21.66). It can be observed that although the decay factor of Ad5-nCoV vaccine is lower than that of mRNA vaccines, the levels of neutralizing antibodies induced by the Ad5-nCoV vaccine are lower than mRNA vaccines both 1 month and 6 months post-vaccination [21]. However, it is well known that mRNA vaccines have been reported to cause myocarditis and other cardiac diseases in children and adolescents [22]. We also observed that two doses of the Sinovac inactivated COVID-19 vaccine led to a decrease in neutralizing antibody levels from 5.87 (4.6–7.7) at 28 days after the second dose to 2.28 (2.1–2.5) at 180 days, representing a 61% decline. This suggests that in terms of neutralizing antibody levels and immune durability, Ad5-nCoV may outperform the Sinovac inactivated COVID-19 vaccine [23]. Furthermore, the study indicates that after 180 days post-vaccination, the decline in binding antibody levels against the BA.4/5 variant was greater than the decline in levels against the original strain. Additionally, a study on the durability of the single-dose adenoviral vector vaccine AZD1222 reported that the vaccine can elicit an antibody response as early as three weeks post-vaccination, which can last for at least three months without waning. The binding antibody levels increased from 27.23 (20.84, 35.56) BAU/mL at 28 days post-vaccination to 83.75 (55.98, 125.31) at eleven weeks, showing a continuous increase in antibody levels. Comparatively, standardized binding antibody levels three months post-vaccination were higher for AZD1222 at 67.7 BAU/mL (95% CI, 49.7–91.9) compared to Ad5-nCoV, and the peak times for the two vaccines were notably different [24]. However, AZD1222 was previously reported to induce a Grade 4 fever of 40.2 °C in a participant aged 6–11 years on the first day post-vaccination, which resolved within 24 h [25]. 

In summary, safe and effective COVID-19 vaccines that can be used for children and adolescents are eagerly anticipated by the public. Due to the prioritization of vaccinating the elderly population in most countries, there has been an increase in the proportion of cases and hospitalizations among school-age children and adolescents compared to the early stages of the pandemic. Additionally, the direct benefits of preventing children from contracting SARS-CoV-2 include averting severe illness, hospitalization, and serious or long-term complications such as MIS-C [17]. The indirect benefits include reducing the likelihood of transmission in home and school environments. Without an effective COVID-19 vaccine for this age group, children could become a persistent source of infection and a breeding ground for new emerging variants [26].

There are some limitations to our study. First, humoral immunity was selected as a measure of non-inferiority and immune consistency in this study, and vaccine-induced T-cell responses were not measured. Secondly, all the study populations are from one site, and the results might not be well representative. In addition, in this study, the children and adolescent participants only included individuals aged 6–17 years, excluding children under 6 years old. Further research is warranted in the future. Fourth, due to the implementation of quarantine policies in China during the clinical trials, there were very few patients, so there was a lack of protective efficacy data with the disease as the observation endpoint. Fifth, due to the relatively small sample size, it is insufficient to determine the potential risks of some rare but severe adverse reactions, such as vaccine-induced immune thrombocytopenia.

Ad5-nCoV still has advantages over all other available COVID-19 vaccines for children and adolescents. Firstly, the single-dose vaccination process is simple and allows for more people to be vaccinated within the same time frame, thus enabling the protection of as many individuals as possible in the shortest amount of time [27]. Secondly, although the proportion of severe cases in children is low, it can help alleviate the pressure on hospital bed capacity within the same timeframe. Secondly, even though the proportion of severe cases in children is low, it helps to reduce severe cases in adults, thereby alleviating the pressure on hospital bed capacity at the same time. Additionally, Ad5-nCoV only requires storage at temperatures of 2–8 °C, making it suitable for transportation and distribution [28].

## 5. Conclusions

The results of this trial confirm that the Ad5-nCoV intramuscular injection has good immunogenicity in the 6–17 age group, with safety comparable to that in adults. This supports the further research and exploration of Ad5-nCoV.

## Figures and Tables

**Figure 1 vaccines-12-00683-f001:**
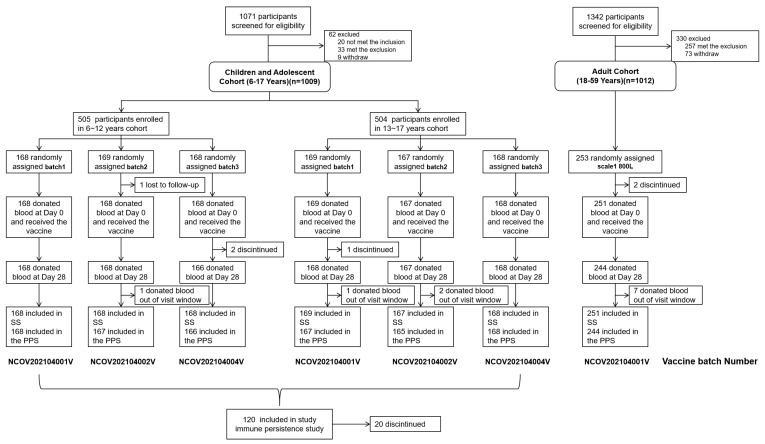
Trial profile. Safety set (SS): All subjects who received vaccination after randomization should be evaluated for safety. Data that violate the protocol should not be culled. Per protocol set (PPS): Subjects in this dataset were more compliant with the protocol, had no significant protocol violations during the study period, and met all inclusion/exclusion criteria.

**Figure 2 vaccines-12-00683-f002:**
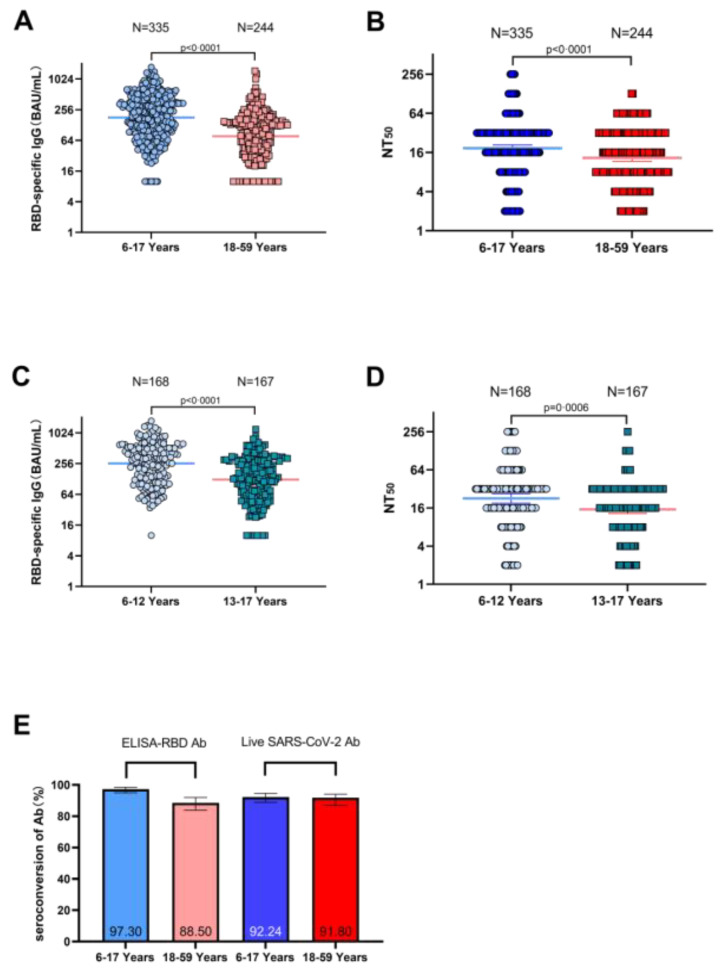
SARS-CoV-2 binding antibodies’ levels on day 28 post-vaccination (**A**). SARS-CoV-2 neutralizing antibodies’ levels on day 28 post-vaccination (**B**) in participants aged 6–17 years and 18–59 years. SARS-CoV-2 binding antibodies’ levels on day 28 post-vaccination (**C**). SARS-CoV-2 neutralizing antibodies’ levels on day 28 post-vaccination (**D**) in participants aged 6–12 years and 13–18 years. Seroconversion of anti-RBD antibodies and neutralizing antibodies on day 28 post-vaccination (**E**) in participants aged 6–17 years and 18–59 years. Error bars are 95% CIs. NT50 = 50% neutralizing titer. FIG. 2ABCD performs a t test on the log-converted data, and FIG. 2E adopts the Chi-square test.

**Figure 3 vaccines-12-00683-f003:**
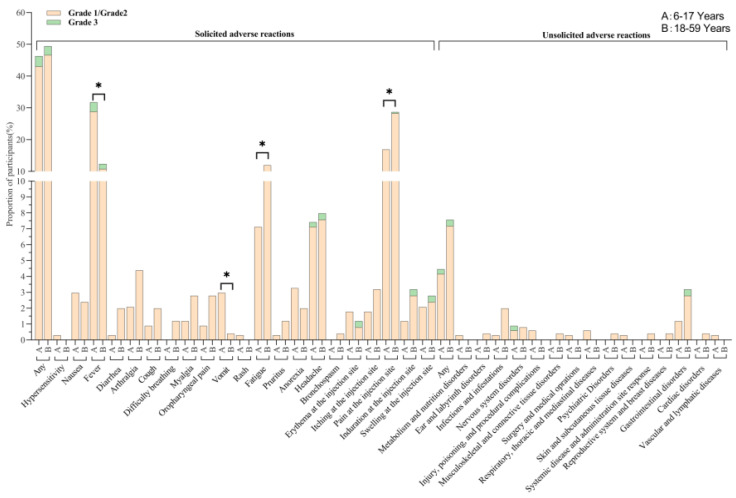
Solicited and unsolicited adverse reactions occurring within 28 days after the vaccination. A = 6–17 years (800L batch 1). B = 18–59 years (800L Scale 1). Any = all the participants with any adverse reactions. The analysis was based on the safety set cohort, with all participants receiving vaccines after randomization. * *p* value < 0.05. Comparisons between groups were made using the Chi-square Test or Fisher’s Exact Test.

**Figure 4 vaccines-12-00683-f004:**
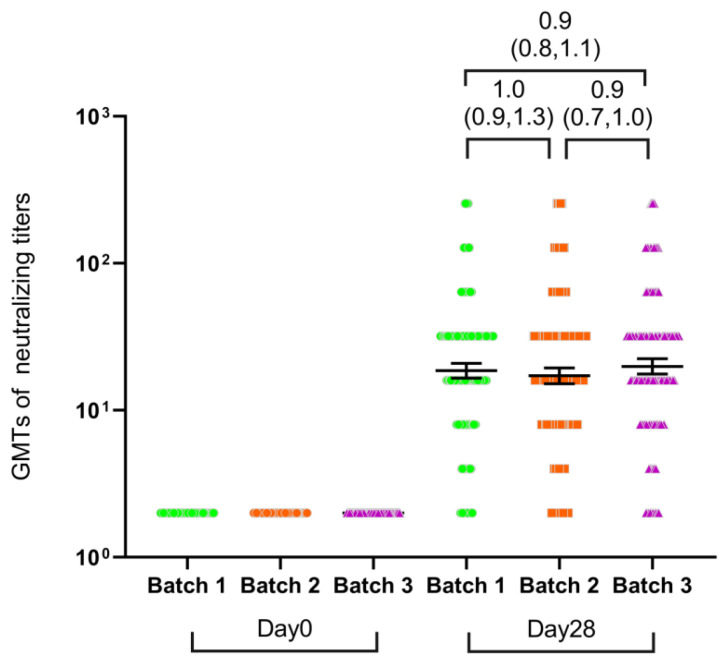
Geometric mean titers (GMTs) of SARS-CoV-2 neutralizing antibodies in three batches of vaccine recipients on day 0 or day 28 post-vaccination. GMT = geometric mean titer, error bars represent two-sided 95% confidence intervals. Brackets above the dots indicate GMT ratios and two-sided 95% CIs between each pair of batches. Inter-group comparisons were made using the ANOVA method of log-converted data.

**Figure 5 vaccines-12-00683-f005:**
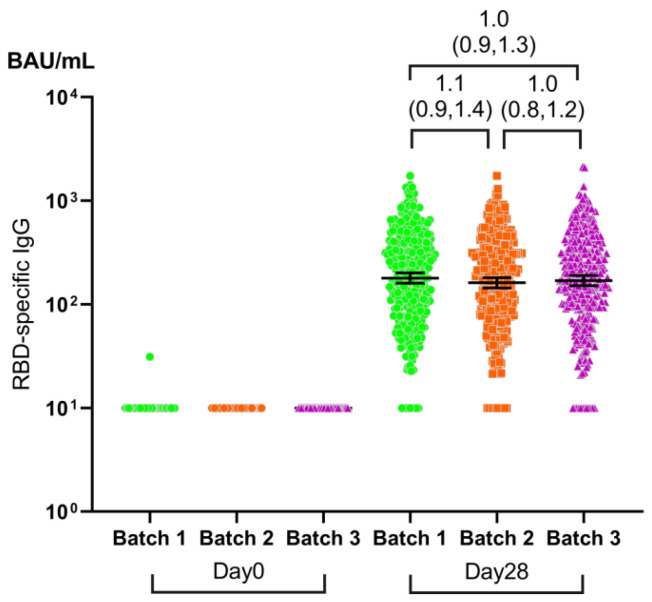
GMC of SARS-CoV-2 RBD-specific IgG on day 0 and 28 post-vaccination. GMC = geometric mean concentration. BAU = binding antibody unit. Error bars are GMC with two-sided 95% CIs. Brackets above the dots indicate GMC ratios and two-sided 95% CIs between each pair of batches. Inter-group comparisons were made using the ANOVA method of log-converted data.

**Figure 6 vaccines-12-00683-f006:**
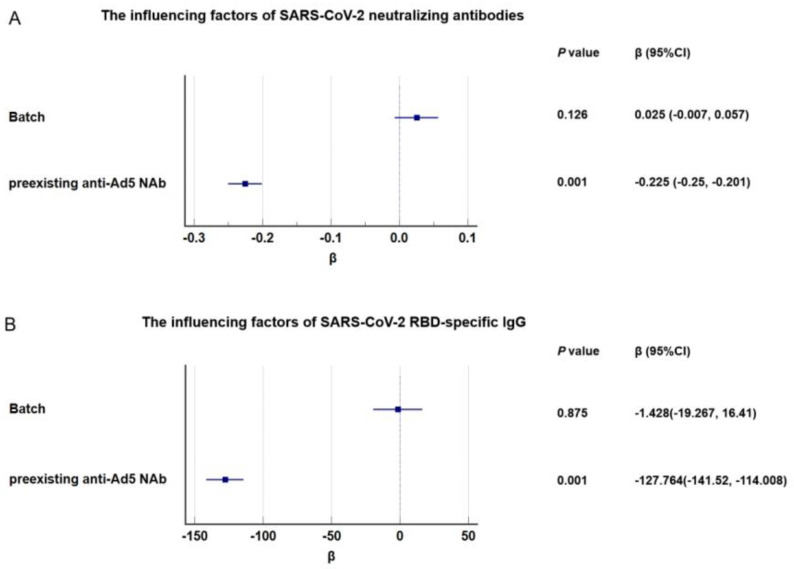
The influencing factors for the SARS-CoV-2 neutralizing antibodies (**A**) and SARS-CoV-2 RBD-specific IgG (**B**) in the 6–17 age cohort.

**Figure 7 vaccines-12-00683-f007:**
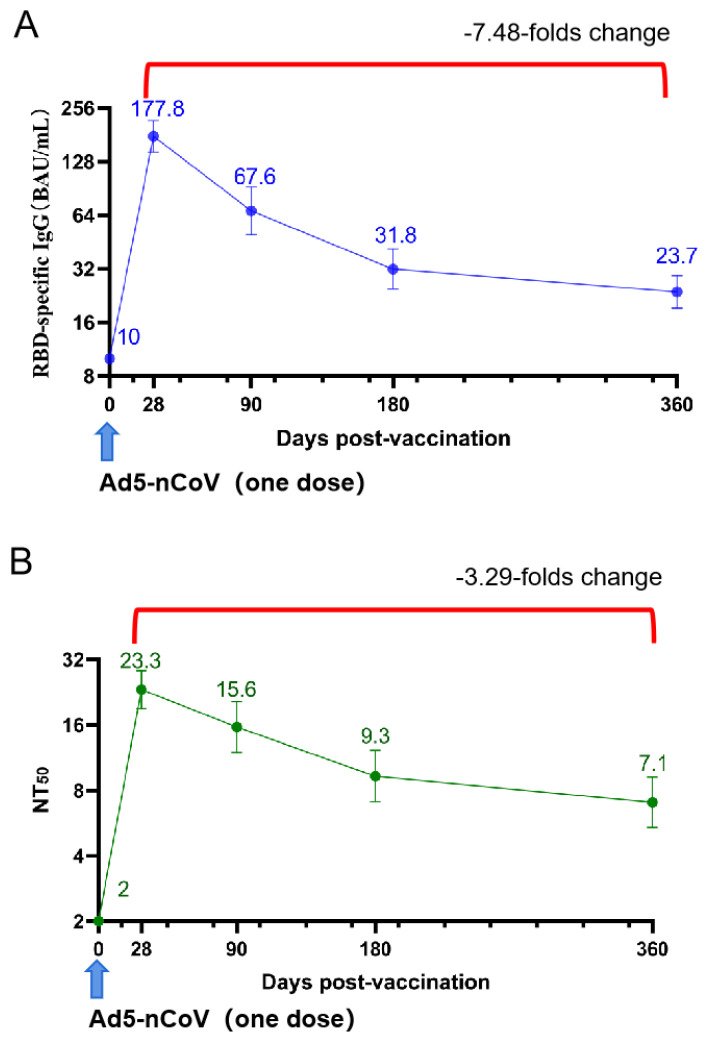
SARS-CoV-2 RBD binding antibodies’ level on the day of vaccination and one year post-vaccination (**A**) and SARS-CoV-2 neutralizing antibodies’ level (**B**) in participants aged 6–17 years. BAU = binding antibody unit. Error bars are GMT/GMC with two-sided 95% CIs. The values in the curly brackets represent the ratio of the peak value to the lowest value after decay.

**Figure 8 vaccines-12-00683-f008:**
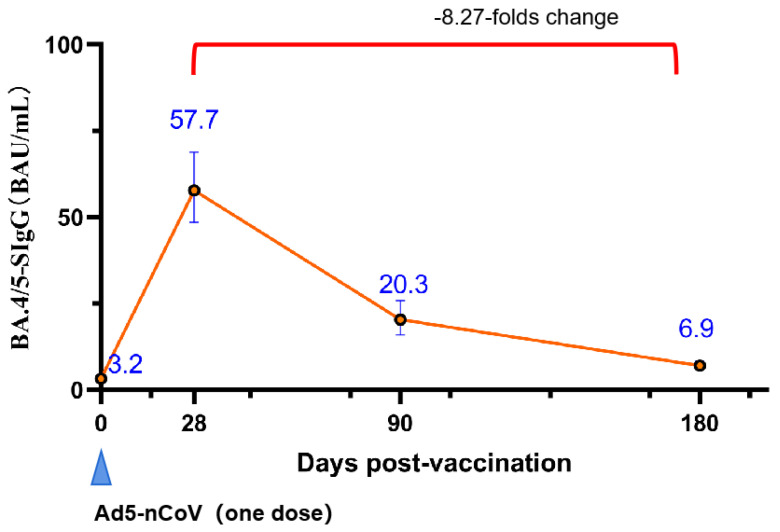
BA.4/5 binding antibodies’ level on the day of vaccination and 180 days post-vaccination in participants aged 6–17 years. BAU = binding antibody unit. Error bars are GMC with two-sided 95% CIs. The values in the curly brackets represent the ratio of the peak value to the lowest value after decay.

**Table 1 vaccines-12-00683-t001:** Baseline demographic data of study cohort.

Assessment Variables	Children and Adolescent Cohort (6–17 Years)			Adult Cohort (18–59 Years)
Analysis Cohort	Batch 1	Batch 2	Batch 3	Scale A-800L
Dug batch Number	NCOV202104001V	NCOV202104002V	NCOV202104004V	NCOV202104001V
Participants in safety analysis	N = 337	N = 335	N = 336	N = 251
Age—no. (%)				
6–12 years	168 (50.1%)	168 (50.1%)	168 (50.0%)	
13–17 years	169 (49.9%)	167 (49.9%)	168 (50.0%)	
Mean—year (mean (SD))	11.93 (2.96)	11.99 (2.81)	11.76 (3.04)	42.13 (9.88)
Sex—no. (%)				
Male	194 (57.6%)	183 (54.6%)	169 (50.3%)	115 (45.82%)
Female	143 (43.4%)	152 (45.4%)	167 (49.7%)	136 (54.18%)
Height (cm)	151.59 (16.00)	151.82 (15.87)	150.88 (16.91)	162.07 (8.20)
Weight (kg)	47.36 (17.33)	47.36 (16.08)	46.74 (17.18)	67.15 (11.39)
BMI (kg/m^2^)	19.97 (4.49)	20.05 (4.47)	19.91 (4.42)	25.53 (3.63)
Participants in immunogenicity analysis	n = 335	n = 332	n = 334	n = 244
Pre-existing NAb against Ad5				
Titer ≤ 200	158 (47.2%)	142 (42.8%)	150 (44.9%)	111(45.5%)
Titer > 200	177 (52.8%)	193 (57.2%)	184 (55.1%)	133 (54.5%)

BMI: body mass index. Data are n (%) or mean (SD), unless otherwise stated. The analysis was based on the Per Protocol Set (PPS), and the subjects in this dataset were more compliant with the protocol, had no significant protocol violations during the study period, and met all inclusion/exclusion criteria.

## Data Availability

The data presented in this study are available upon request from the corresponding author.

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
