# Peer review of "Immunogenicity, Safety, and Immune Persistence of One Dose of SARS-CoV-2 Recombinant Adenovirus Type-5 Vectored Vaccine in Children and Adolescents Aged 6–17 Years: An Immunobridging Trial"

_vaccines, 2024, doi:10.3390/vaccines12060683_

Round 1

Reviewer 1 Report

Comments and Suggestions for Authors

The topic is clinically relevant and interesting.
Some comments regarding statistics and study design:

1. In my opinion, it is not appropriate to call this study a "randomized trial". Although the participants are randomly assigned to one of three batches, it is not the aim and the purpose of this study to compare these batches.

2. When performing a non-inferior test, a inferiority margin should be predefined.

3. Section 2.7: The epxpression "to test for normaly data" is not correct. For instance, you could formulate: Shapiro-Wilk test was used to check whether the sample stems from a normally distributed population. - For safety parameters (Section 3.2.2) Chi2 test or Fisher's exact test has been applied. Please indicate this in section 2.7.

Results, line 21: 1001/1008 = 99.3% (not 99,2%).

Results, table 1:
For batch 1, percentage values have been swapped (168 = 49.9%, 169 = 50.1%).
Ages are presented by mean ± standard deviation; howver weight and BMI are given by mean and range. This is inconsistent. I think you should oresent mean ± standard deviation for all these variables. 

Figure 2E: The term "whisker" shoukd be avoided for this kind of diagram (this is not a Box-and-Whisker-Plot").  The geometric mean titers are given by the boxes. 

Line 250: The percentages (97.3% and 88.5%) are not evident from the figure. 2C.

Line 251: "The levels .. are similar (p = 0.073)". Significance was only narrowly missed. Therefore, it may be informative to present the levels of the two cohorts in order to enable a direct comparison.

3.2.2 Safety: Why are there only 337 participants in the adolescent group?

Line 291: "Multiple regression was performed ..." Which explanatory vraibles have been considered when performing a multiple regression analysis?

In gerenal: When did you use t test, Wilcoxon 2 sample test or ANOVA? Please indicate this in the relevant text passages.

Comments on the Quality of English Language

I think the quality of the english language is fine.
Line 397: The dot should be removed.

Author Response

Thank you very much for the comments. The following is a detailed reply to the review comments.

Review1

Comment 1: In my opinion, it is not appropriate to call this study a "randomized trial". Although the participants are randomly assigned to one of three batches, it is not the aim and the purpose of this study to compare these batches.

Response: The reviewer believe that although the subjects were randomly assigned to three different batches in this study, the main purpose of this study is to highlight the bridging trial. I also agree that improvements are needed, so the title has been revised to "Immunogenicity, Safety and Immune Persistence of One Dose of SARS-CoV-2 Recombinant Adenovirus type-5 Vectored Vaccine in Children and Adolescents Ages 6-17 Years: An Immunobridging Trial".

Comment 2: When performing a non-inferior test, a inferiority margin should be predefined.

Response: The inferiority margin was mentioned in the statistical section at line 160 of the article, where the geometric mean ratio (GMR) was reported as 0.67. To clarify this, a statement regarding the inferiority margin was added in the statistical section as follows: "Non-inferiority was defined as the lower limit of the two-sided 95% confidence interval of a GMR > 0.67." Please refer to the revisions made on line 180 of the manuscript.

Comment 3: Section 2.7: The epxpression "to test for normaly data" is not correct. For instance, you could formulate: Shapiro-Wilk test was used to check whether the sample stems from a normally distributed population. - For safety parameters (Section 3.2.2) Chi2 test or Fisher's exact test has been applied. Please indicate this in section 2.7.

Response: Based on the reviewer's comments, corrections have been made to the statistical analysis section in lines 182 and 186 of the article.

Comment 4: Results, line 21: 1001/1008 = 99.3% (not 99,2%).

Response: This error was corrected on line 210 of the article

Comment 5: Results, table 1:

For batch 1, percentage values have been swapped (168 = 49.9%, 169 = 50.1%).

Ages are presented by mean ± standard deviation; howver weight and BMI are given by mean and range. This is inconsistent. I think you should oresent mean ± standard deviation for all these variables.

Response: In the Table 1 section, the representation of height and weight has been changed to mean (standard deviation). 

Comment 6: Figure 2E: The term "whisker" shoukd be avoided for this kind of diagram (this is not a Box-and-Whisker-Plot"). The geometric mean titers are given by the boxes.

Response: In Fig2, the wording issue has been amended to "Error bars are 95% CIs."

Comment 7: Line 250: The percentages (97.3% and 88.5%) are not evident from the figure. 2C.

Response: Thank you very much for the reviewer's suggestions. There was error in the expression of this article, where 97.3% and 88.5% were shown in Figure 2E. These have now been corrected. Please refer to the revisions made on line 247 of the manuscript.

Comment 8: Line 251: "The levels .. are similar (p = 0.073)". Significance was only narrowly missed. Therefore, it may be informative to present the levels of the two cohorts in order to enable a direct comparison.

Response: The statement "The levels .. are similar (p = 0.073)" has now been specified with actual numerical values in the text. Please refer to the revisions made on line 249 of the manuscript.

Comment 9: 3.2.2 Safety: Why are there only 337 participants in the adolescent group?

Response: For the safety analysis, the number of children and adolescents utilized in the bridging portion of this study was 337 individuals, consistent with the immunogenicity study in the bridging trial. All children and adolescent participants included in this study will be included in the analysis for batch-to-batch consistency assessment.

Comment 10: Line 291: "Multiple regression was performed ..." Which explanatory vraibles have been considered when performing a multiple regression analysis?

Response: Due to this statistical verification being specifically focused on batch consistency, there were no differences in age and gender among the groups, so only batches and preexisting anti-Ad5 NAb were included.

Comment 11: In gerenal: When did you use t test, Wilcoxon 2 sample test or ANOVA? Please indicate this in the relevant text passages.

Response: The specific analytical methods used for each analysis have been added according to the opinion of the review experts and are highlighted below the picture.

Reviewer 2 Report

Comments and Suggestions for Authors

In the manuscript entitled “Immunogenicity, Safety and Immune Persistence of One Dose of SARS-CoV-2 Recombinant Adenovirus type-5 Vectored Vaccine in Children and Adolescents Ages 6-17 Years: A Single Center, Randomized, Double-Blind, Immunobridging Trial”, the authors investigated both the safety and effectiveness of a single-dose Ad5-nCoV COVID-19 vaccine in children (aged 6-12 years) and adolescents (aged 13-17 years) compared to adults (aged 18-59 years). This single-centre, randomized double-blinded, immunobridiging study found a single dose of Ad5-nCoV safe in children and adolescents. The rate of side effects was similar between the children/adolescent group and the adult group. The vaccine caused a significant immune response in children and adolescents. Interestingly, the immune response was not only comparable to, but even enhanced than, the response shown in adults.

The manuscript is well-written and straightforward. The information flow is clear. I have the following comments:

1.       “The trial has been registered with ClinicalTrials.gov, registration 190 number NCT04916886.” It shouldn’t be in the abstract or statistical analysis section (2.7). It is already in the study design section (2.1)

2.       Figure 1 is hazy, please enhance the figure quality.

3.       The study is a single centre, investigation in other centres is also recommended to confirm the consistency of the results, as authors mentioned in the discussion.

4.       Measuring the immune persistence beyond one year is also recommended. Figures 7 and 8 show a gradual decline in antibody levels over time. This will show when the booster is required.

5.       One of the limitations here is the testing of efficacy against one variant, BA.4/5 omicron variants (up to July 2021). I would recommend testing efficacy against current or common emerging variants.

6.       Ad5-nCoV is advocated as a single-dose vaccine and booster. Hence, it is recommended to have future research to compare the safety and efficacy of the drug between children and adolescents versus adults after booster dose.

Overall, it's an interesting manuscript and scientifically sound.

Author Response

Thank you very much for the comments. The following is a detailed reply to the review comments.

Review2

Comment 1: “The trial has been registered with ClinicalTrials.gov, registration 190 number NCT04916886.” It shouldn’t be in the abstract or statistical analysis section (2.7). It is already in the study design section (2.1)

Response: The ClinicalTrials registration number in the abstract section and the Statistics section (2.7) has been removed.

Comment 2: Figure 1 is hazy, please enhance the figure quality.

Response: Figure1 has been changed to a cleaner version. Please refer to the revisions made on line 203 of the manuscript.

Comment 3: The study is a single centre, investigation in other centres is also recommended to confirm the consistency of the results, as authors mentioned in the discussion.

Response: One limitation of this paper is that it is a single-center study. Thanks for the comments from the reviewers. We are also striving to promote the research of other centers.

Comment 4: Measuring the immune persistence beyond one year is also recommended. Figures 7 and 8 show a gradual decline in antibody levels over time. This will show when the booster is required.

Response: As suggested by the review experts, the follow-up research on antibody tracking and booster injection is also under continuous research, which will be shown in subsequent articles.

Comment 5: One of the limitations here is the testing of efficacy against one variant, BA.4/5 omicron variants (up to July 2021). I would recommend testing efficacy against current or common emerging variants.

Response: BA.4/5, a new strain that emerged in May 2022, is spreading faster than other Omicron variants and may cause more hospitalizations and deaths. Therefore, this relatively representative strain was selected for detection in this study. In addition, at this time, we collected serum from all subjects for the immune persistence study test, so we chose the prevailing strain at that time. Of course, this study has several limitations. In future research, we will expand the detection of variant strains to provide a more comprehensive analysis. For this study, we will include this point in the limitations section. Please refer to the revisions made on line 496 of the manuscript.

Comment 6: Ad5-nCoV is advocated as a single-dose vaccine and booster. Hence, it is recommended to have future research to compare the safety and efficacy of the drug between children and adolescents versus adults after booster dose.  

Response: As suggested by reviewer, Ad5-nCoV is advocated as either a single-dose vaccine or a booster. Therefore, we hope that in the future, we can also compare Ad5-nCoV as a booster shot and examine the differences between children and adolescents versus adults . Currently, this point is listed under limitations, and future research will consider this comparison. Please refer to the revisions made on line 489 of the manuscript.

Round 2

Reviewer 1 Report

Comments and Suggestions for Authors

Thank for having considered and commented for having considered and commented on all my suggestions. The paper is very fine now.

Author Response

Thank you again for your feedback and affirmation.